# Study protocol for a randomised trial for atosiban versus placebo in threatened preterm birth: the APOSTEL 8 study

Job Klumper ![ORCID],[1] Wouter Breebaart,[1] Carolien Roos,[1] Christiana A Naaktgeboren,[2] Joris van der Post,[1] Judith Bosmans,[3] Anton van Kaam,[4] Ewoud Schuit,[2,5] Ben W Mol,[6] Jelle Baalman,[7] Fionnuala McAuliffe,[7,8] Jim Thornton,[9] Marjolein Kok,[1] Martijn A Oudijk ![ORCID] [1]

For numbered affiliations see end of article.

**Correspondence to**
Dr Martijn A Oudijk;
m.a.oudijk@amsterdamumc.nl

## ABSTRACT

**Introduction** Preterm birth complicates >15 million pregnancies annually worldwide. In many countries, women who present with signs of preterm labour are treated with tocolytics for 48 hours. Although this delays birth, it has never been shown to improve neonatal outcome. In 2015, the WHO stated that the use of tocolytics should be reconsidered and that large placebo-controlled studies to evaluate the effectiveness of tocolytics are urgently needed.

**Methods and analysis** We designed an international, multicentre, randomised, double-blinded, placebo-controlled clinical trial. Women with threatened preterm birth (gestational age 30–34 weeks), defined as uterine contractions with (1) a cervical length of < 15 mm or (2) a cervical length of 15–30 mm and a positive fibronectin test or (3) in centres where cervical length measurement is not part of the local protocol: a positive fibronectin test or insulin-like growth factor binding protein-1 (Actim-Partus test) or (4) ruptured membranes, will be randomly allocated to treatment with atosiban or placebo for 48 hours. The primary outcome is a composite of perinatal mortality and severe neonatal morbidity. Analysis will be by intention to treat. A sample size of 760 participants (380 per group) will detect a reduction in adverse neonatal outcome from 11.95% to 6% (alpha error 0.05, beta error 0.2). A cost-effectiveness analysis will be performed from a societal perspective.

**Ethics and dissemination** This study has been approved by the Research Ethics Committee (REC) of the Amsterdam University Medical Centres, location AMC, as well as the REC's in Dublin and the UK. The results will be presented at conferences and published in a peer-reviewed journal. Participants will be informed about the results.

**Trial registration number** Nederlands Trial Register (Trial NL6469).

## Strengths and limitations of this study

⇒ The primary outcome is perinatal mortality and neonatal morbidity, not prolongation of pregnancy.
⇒ This is the largest randomised trial comparing atosiban to placebo for women with threatened preterm birth.
⇒ Over 40 hospitals in Europe will participate.
⇒ Tocolysis is incorporated in daily routine as it has been the recommendation in many guidelines.
⇒ It will prove to be a challenge in counselling patients to participate in a placebo controlled trial, especially in an acute setting.

## INTRODUCTION

Preterm birth, defined as birth before 37 weeks' gestation, is a major contributor to perinatal mortality and morbidity, complicating over 15 million pregnancies worldwide.[1 2] Of all infant deaths before the age of 5 years, more than one-third can be attributed to preterm birth.[3] In addition, spontaneous preterm birth is the leading cause of neonatal morbidity, mostly due to respiratory immaturity, intracranial haemorrhage and infections.[4 5] These conditions can have long-term neurodevelopmental sequelae such as cognitive impairment, cerebral palsy and visual and hearing deficiencies. Preterm birth is one of the largest single contributors to the global burden of disease because of the high mortality early in life and the morbidity of lifelong impairment.[6]

Maternal administration of corticosteroids to accelerate fetal lung maturation is an effective treatment for women with threatened preterm birth.[7] Since steroids have their maximum effect if birth is delayed by 48 hours, many obstetricians administer a tocolytic drug alongside the steroids to allow maximal steroid effect and facilitate transport of the mother to a centre with neonatal intensive care unit facilities if needed. Several tocolytics are used, including β adrenoceptor agonists, cyclooxygenase inhibitors (COX), magnesium sulfate, calcium-channel blockers and oxytocin receptor antagonists. Though more or less effective in delaying delivery, no tocolytics used in obstetrical practice are proven effective in reducing neonatal

morbidity and mortality.[8 9] None of the studies so far have been powered to show such an effect.

The two most commonly used tocolytic drugs, atosiban and nifedipine, showed comparable perinatal outcome in the APOSTEL 3 study.[10] However, neonatal mortality was higher in the nifedipine group, although not significant (5.4% vs 2.4% relative risk (RR) 2.20; 95% CI 0.91 to 5.33).

The oxytocin receptor antagonist atosiban has fewer maternal side effects in head to head comparison with alternative drugs,[11] and showed similar effectiveness in delaying birth compared with ritodrine.[12] In placebo-controlled trials, a Cochrane review showed that atosiban did not reduce perinatal mortality (RR 2.25, 95% CI 0.79 to 6.38; two studies with 729 infants) or major neonatal morbidity,[13] although the quality of this review has been questioned.[14]

One explanation might be that since spontaneous preterm birth is associated in 40%–70% of cases with chorioamnionitis,[15 16] tocolysis may prolong fetal exposure to an infectious environment, which may worsen neonatal outcome.

Perinatal outcome has also markedly improved over the last few decades, in part due to postnatal interventions such as exogenous surfactant treatment which reduces mortality and respiratory morbidity in preterm infants.[17] This might also limit the potential benefit of tocolytics.

Worldwide, practice varies widely. Several large institutions in countries like Canada, Scotland and Ireland, rarely use tocolytics, while in the USA, COX (indomethacin) and calcium channel blockers (nifedipine) are popular. In Europe, nifedipine and the oxytocin antagonist, atosiban, are both widely used.

In conclusion, current widespread use of tocolytic drugs for this indication is not supported by the available evidence. The primary goal of tocolysis should not be prolongation of pregnancy, but improvement of neonatal outcome. This view is supported by the WHO, as they state in their 2015 guidelines on preterm birth that the effectiveness of tocolytics is not proven, and that placebo-controlled studies are urgently needed.[18] Based on the results of the APOSTEL 3 study,[10] the associated editorial,[19] and its safety profile we chose to evaluate atosiban in the APOSTEL 8 study.

### Objective
To test the hypothesis that tocolysis with atosiban in late preterm birth (30–34 weeks) reduces neonatal mortality and morbidity and is cost-effective compared with placebo.

## METHODS AND ANALYSIS
### Design and setting
We will conduct an international, multicentre, double-blind, randomised, placebo-controlled clinical trial, performed in The Netherlands, UK and Ireland.

### Participants/eligibility criteria
Women, aged ≥18 years, with threatened preterm birth and a gestational age between $30^{+0}$ and $33^{+6}$ weeks are eligible. Threatened preterm birth is defined as uterine contractions with

1. A cervical length of <15 mm or
2. A cervical length of 15–30 mm and a positive fibronectin test or
3. In centres where cervical length measurement is not part of the local protocol: a positive fibronectin test or insulin-like growth factor binding protein-1 (Actim-Partus test) or
4. Ruptured membranes.

These inclusion criteria are based on the results and conclusions of the APOSTEL 1 study[20] and current guidelines within the Netherlands and the UK. Moreover, our previous APOSTEL 3 study, with resembling inclusion criteria, showed that half of the women with these criteria deliver within 7 days,[10] validating this definition of women at high risk for preterm birth. In addition, the sample size of expected adverse neonatal outcome in the gestational age group of 30–34 weeks, was calculated from the APOSTEL 3 study.

This study was designed in a pragmatic fashion, in order for the results to be applicable in the current clinical practice. As most national guidelines and local protocols propose treatment for threatened preterm birth in both singleton and multiple pregnancies, as well as women with ruptured membranes, all these categories of patients are eligible for the study.

Women with a contraindication for tocolysis, signs of fetal distress, clinical signs of intrauterine infection, previous treatment for threatened preterm birth with corticosteroids in the current pregnancy and known fetal chromosomal or severe structural abnormalities are not eligible.

### Procedures, recruitment, randomisation and collection of data
Potential participants will be identified by the local research co-ordinators and/or the staff of participating hospitals. Women eligible for the trial will be counselled by doctors, midwifes or research nurses trained in 'good clinical practice', and will be given a patient information form to read. Those who wish to participate, will be asked to give for written informed consent and are registered within the central trial database. Randomisation will be performed by using sequentially numbered medication packs available in each centre. Only the independent data manager has access to the computer-generated randomisation list in which the medication numbers are linked with atosiban or placebo. Treatment allocation is blinded to investigators, participants, clinicians and research coordinators. Randomisation will be balanced with varying block sizes of 2 and 4, and stratified by centre.

At study entry, baseline demographic, prior obstetric and medical history will be recorded into the web-based case report form accessible through a secure central website (Castor Electronic Data Capture, Ciwit B.V.).[21]

Details of delivery, maternal and neonatal assessments during pregnancy and postpartum period will be recorded on the same system. All data will be coded, processed and stored with adequate precautions to ensure patient confidentially. This is described in a separate data management plan.

## Interventions

Participants are allocated to atosiban or matching placebo (0.9% saline) for 48 hours. The medication will be administered by a bolus injection of 6.75 mg/0.9 mL in 1 min followed by a continuous infusion of 18 mg/hour for 3 hours followed by a continuous infusion of 6 mg/hour for the remaining 45 hours. Participating women will otherwise be treated according to local protocol based on national guidelines, including corticosteroids MgSO$_4$ for neuroprotection and antibiotics if needed.

## Outcome measures

Outcome parameters are in line with the core outcome set for studies on prevention of preterm birth defined by members of GONet and the Core Outcomes in Women's health (CROWN) initiative (www.crown-initiative.org).[22]

The primary outcome measure is a composite of adverse perinatal outcome composed of perinatal in-hospital mortality and six severe perinatal morbidities: bronchopulmonary (BPD), periventricular leucomalacia (PVL) >grade 1, intraventricular haemorrhage >grade 2, necrotising enterocolitis (NEC) ≥stage 2, retinopathy of prematurity >grade 2 or needing laser therapy, and culture proven sepsis.

The diagnosis of BPD will be made according to the international consensus guideline as described by Jobe and Bancalari.[23] PVL >grade 1 and intraventricular haemorrhage >grade 2 will be diagnosed by repeated cranial ultrasound according to the guidelines on neuroimaging described by de Vries et al[24] and Ment et al.[25] NEC ≥stage 2 will be diagnosed according to Bell et al.[26] Culture proven sepsis is diagnosed on the combination of clinical signs and positive blood cultures. The components of the composite adverse perinatal outcome will also be assessed separately.

Secondary infant outcomes will be birth within 48 hours, time to birth, gestational age at birth, birth weight, number of days on invasive mechanical ventilation, length of neonatal intensive care unit (NICU) stay, convulsions, asphyxia, meningitis, pneumothorax until hospital discharge.

Maternal outcomes will be mortality, infection of inflammation and harm to mother from interventions (side effects). Side effects are defined as admission to intensive care, anaphylactic shock, dyspnoea, hypotension (leading tocardiotocography abnormalities), liver test abnormalities (elevated aspartate aminotransferase (ASAT) or alanine aminotransferase (ALAT)), general side effects (nausea, vomiting, headache), postpartum haemorrhage defined as >500 mL blood loss and maternal mortality.

We will ask informed consent to approach the parents for long-term follow-up of the children. We intend, subject to funding, to use standardised questionnaires at 2 and 5 years of age.

Maternal quality of life will be assessed at randomisation and at 3 months baby corrected age using the EuroQol (EQ-5D-5L) questionnaire. This consists of five dimensions (mobility, self-care, usual activities, pain/discomfort, anxiety/depression) that are rated using five levels (no problems, slight problems, moderate problems, severe problems, extreme problems).

Societal costs will be assessed using adapted versions of the iMTA Medical and Productivity Cost Questionnaires at 3 months corrected baby age. Cost data include costs of the intervention, other healthcare utilisation, patient and family costs and costs of productivity losses.

## Withdrawal of subjects

Participants can cease study treatment at any time for any reason if they wish to do so. Unless they refuse to allow further data collection, such participants will continue to be followed-up and will be analysed in the group to which they were originally allocated. Participants who decline follow-up will have no further trial data collected. Any results collected up to the point at which they decline follow-up will be analysed. Study medication will be discontinued in patients with signs of intrauterine infections or signs of fetal distress (abnormal CTG, meconium stained amniotic fluid). Data of such participants will continue to be analysed. Further management will be left to the expertise of the responsible clinician. The responsible clinician can contact a perinatologist from the project group in case of suspected side effects or other medical problems. If necessary, treatment will be discontinued.

## Monitoring and safety

An independent data safety monitoring board (DSMB) will focus on both effectiveness and safety. Serious adverse events (SAEs) will be collected from the first study-related procedure until 3 months after delivery. All SAEs will be reported to the DSMB within 7 days. Whenever there is proof of effectiveness (at interim analysis) or safety issues (increased (serious) adverse events in one of the two treatment arms) the DSMB will advise whether the trial should be stopped or continued. The DSMB will be blinded when first analysing the data, but unblinded before reaching a decision.

The advice(s) of the DSMB will only be sent to the sponsor of the study. Should the sponsor decide not to fully implement the advice of the DSMB, the sponsor will send the advice to the reviewing Medical Ethics Committee, including a note to substantiate why (part of) the advice of the DSMB will not be followed.

A formal interim analysis is planned after data collection of 500 women. At these interim analyses, the Haybittle-Peto alpha spending function will be used, which means that an effect at interim with a p value <0.001 is considered statistically significant.

## Sample size

Based on the APOSTEL 3 data, the proportion of adverse perinatal outcome in women randomised between 30 and 34 weeks gestation and treated with atosiban was 6%.[10] Based on two recent studies,[27 28] we expect a 49,8% reduction of 11.95% adverse perinatal outcome in the placebo group to 6% in the atosiban group. Therefore we need to randomise 722 women (beta-error 0.2; alpha error 0.05). Assuming a 5% drop-out rate, we need to randomise 760 women (380 in each arm).

## Statistical analysis

### Data analysis

Data analysis will be performed according to the intention-to-treat principle. In the baseline table, categorical variables will be expressed as a number with the percentage of the total allocation arm. Continuous variables will be presented as mean with SD, as geometric mean with 95% CI or as median with IQR, whichever appropriate.

The main outcome 'adverse neonatal outcome' will be assessed on the infant level, using a log-binomial generalised estimating equations model to take into account the correlation of outcomes in multiples, resulting in an RR with accompanying 95% CI. To account for stratified randomisation by centre, we will also take centre into the model if the model converges.[29] We will account for interdependence between outcomes in multiple pregnancies by considering the mother as a cluster variable.[30]

The other secondary outcome measures on the child level will be analysed similar to the primary outcome measure. Outcomes on the maternal level will be assessed by using a binomial regression model with log-link function. When a statistically significant difference in primary outcome is found between both groups, we will calculate the number needed to treat. Time to delivery will be evaluated by Kaplan-Meier estimates and Cox proportional hazard analysis, taking into account the different durations of gestation at study entry, and will be tested with the log rank test. A p value of <0.05 will be considered to indicate statistical significance.

### Subgroup analyses

The following subgroup analyses are planned:
1. Singleton versus multiple pregnancy.
2. Cervical length <15 mm versus cervical length 15–30 mm and a positive fibronectin test (or no cervical length measurement and a positive fibronectin test or Partus test).
3. Ruptured or unruptured membranes at entry.
4. Previous preterm birth.

### Sensitivity analysis

A sensitivity analysis will be performed excluding multiple pregnancies and pregnancies complicated by preterm premature rupture of membranes.

To assess whether a subgroup effect is present, we will add an interaction term between the subgrouping variables and the treatment allocation to the regression model. When an interaction term is statistically significant (p<0.05), we will estimate the treatment effect within strata of the subgrouping variable.

Details of the statistical analysis will be describes in separate statistical analysis plan that will be completed before data lock.

## Cost-effectiveness analysis

The cost-effectiveness analysis will be done according to the intention-to-treat principle. Missing cost and effect data will be imputed using multiple imputation according to the Multivariate Imputation by Chained Equations (MICE) algorithm developed by Buuren.[31] Rubin's rules will be used to pool the results from the different multiply imputed datasets. Bivariate regression analyses will be used to estimate cost and effect differences between atosiban and placebo while adjusting for confounders if necessary. Incremental cost-effectiveness ratios (ICERs) will be calculated by dividing the difference in the mean total costs between the treatment groups by the difference in mean effect between the treatment groups. Bias-corrected and accelerated bootstrapping with 5000 replications will be used to estimate statistical uncertainty. Uncertainty surrounding ICERs will be graphically presented on cost-effectiveness planes. Cost-effectiveness acceptability curves will be estimated showing the probability that atosiban is cost-effective in comparison with placebo for a range of different ceiling ratios thereby showing decision uncertainty. Sensitivity analyses will be performed to assess the robustness of the results using different assumptions regarding costs and effects.

## Patient and public involvement

The preterm birth research line of the Dutch consortium is in close collaboration with two Dutch patient associations, the Vereniging van Ouders van Couveusekinderen (VOC, freely translated to society of parents of children admitted to NICU) and the Nederlandse Vereniging van Ouders van Meerlingen (freely translated to Dutch society of parents of multiples). They are involved in the design of new studies, updated on progress of running trials and informed of study results. Project members are invited speakers at yearly conferences of these societies to present on the progress of our preterm birth research line. At these conferences, surveys are being performed on patient preferences on study ideas. Tocolysis was deemed an important research issue. Both associations have written support letters to the funding agency ZonMw (The Netherlands organisation for health research and development) for the APOSTEL 8 study.

A project panel of parents who experienced a spontaneous preterm birth consisting of six couples was involved in the design of our study. A survey was performed during the design of the study among members of the closed Facebook group of the VOC, to address questions on whether they would be interested in participation in the APOSTEL 8 study.

The Dutch consortium has a website where it publishes all results of completed studies, and publishes the protocols of currently recruiting studies.

Presentations will be held at yearly conferences at patient organisations and updates on research are being published in the journal of the VOC.

## Ethics and dissemination

The Research Ethics Committee (REC) at the Amsterdam University Medical Centres, location AMC, approved this study. Additional regional approval was obtained for the remaining participating hospitals in The Netherlands. Furthermore, the study was approved by the REC of the National Maternity Hospital in Dublin, Ireland, and the REC of East Midlands—Derby in the UK. Protocol amendments will be communicated to a relevant parties. This trial is registered with the Nederlands Trial Register.

A manuscript with the results of the primary study will be published in a peer-reviewed journal. A separate manuscript will be written on the cost-effectiveness analysis. The results of this clinical trial will be presented at conferences and disseminated through publication in a peer-reviewed journal.

### Author affiliations

[1]Department of Obstetrics and Gynaecology, Amsterdam UMC, Location AMC, Amsterdam, The Netherlands
[2]University Medical Centre Utrecht, Julius Center for Health Sciences and Primary Care, Utrecht, The Netherlands
[3]Department of Health Sciences, Amsterdam Public Health Research Institute, Vrije Universiteit Amsterdam, Amsterdam, The Netherlands
[4]Department of Neonatology, Amsterdam UMC, Location AMC and VUmc, Amsterdam, The Netherlands
[5]Standford Prevention Research Center, Stanford University, Stanford, UK
[6]Department of Obstetrics and Gynaecology, School of Medicine, Monash University, Melbourne, Victoria, Australia
[7]Department of Obstetrics and Gynaecology, National Maternity Hospital, Dublin, Ireland
[8]UCD Perinatal Research Centre, University College Dublin, Dublin, Ireland
[9]Department of Obstetrics and Gynaecology, University of Nottingham, Nottingham, UK

**Acknowledgements** We would like to thank the collaborators of the study group; the gynecologists of the participating centres for their help as local investigators for the APOSTEL 8 study. We also thank the patient associations VOC and NVOW for their input in this study.

**Contributors** CAN, JvdP, JB, AvK, ES, BWM, JeB, FM, JT, MK and MAO were involved in conception and design of the study. JK and WB drafted the manuscript. CR, CAN, JvdP, JB, AvK, ES, BWM, JeB, FM, JT, MAO and MK reviewed and edited the manuscript. All authors mentioned in the manuscript are member of the APOSTEL study group or collaborators. They participated in the design of the study during several meetings and are local investigators in the participating centres. All authors edited the manuscript and read and approved the final manuscript

**Funding** This study is funded by ZonMw (The Netherlands organisation for health research and development), grant number 848041004 and the United Kingdom National Institute for Health Research, Clinical Research Network.

**Competing interests** JT received a number of lectures and medical advisory fees from Ferring pharmaceuticals between 2000 and 2016.

**Patient consent for publication** Not required.

**Provenance and peer review** Not commissioned; externally peer reviewed.

**ORCID iDs**
Job Klumper http://orcid.org/0000-0002-2156-8557
Martijn A Oudijk http://orcid.org/0000-0001-8672-4365

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
