## [Reviewer comments · BMJ Open]

ARTICLE DETAILS

TITLE (PROVISIONAL)	Study protocol for a randomised trial for atosiban versus placebo in threatened preterm birth: the APOSTEL 8 study
AUTHORS	Klumper, Job; Breebaart, Wouter; Roos, Carolien; Naaktgeboren, CA; van der Post, Joris; Bosmans, J; van Kaam, Anton; Schuit, Ewoud; Mol, Ben W; Baalman, Jelle; McAuliffe, Fionnuala; Thornton, Jim; Kok, Marjolein; Oudijk, Martijn A

VERSION 1 – REVIEW

REVIEWER	Tabata Zumpano Dias University of Campinas, Brazil
REVIEW RETURNED	22-Jan-2019

GENERAL COMMENTS	It was defined threatened preterm birth (PTB) as women with contractions and 4 strong risk factors for premature delivery. I understand these 4 risk factors as strong risk factors for premature delivery within 7 or 14 days but not necessarily 48 hours, that is the time that tocolysis is performed to allow the complete corticosteroid cycle. As an obstetrician and researcher of spontaneous prematurity (my PhD was on tocolysis and corticosteroid in preterm labor in the largest study on prematurity in my country) I do not see why performing tocolysis if the woman has her cervix closed or if there is no modification of the cervix. I believe that many of the cases that could be included in the sample are cases that would benefit from the corticosteroid without tocolysis. In this way I think that the evaluation of the dilation / modification of the cervix is fundamental to define the need to use tocolysis. Without this evaluation of cervix I think the study does not make sense, so my option to reject (would be a change in methodology), despite understanding how important studies like this (randomized placebo controlled trials) are, mainly with a topic that is controversial (tocolysis) .
--

REVIEWER	Metha Songthamwat Department of Obstetric and Gynecology, Udonthani Hospital, Udonthani, Thailand.
REVIEW RETURNED	28-Jan-2019

GENERAL COMMENTS	Thank you very much for the opportunity to review this valuable protocol. My comments are listed below and a detail's file is attached to the editors and author -Title 1.Why do the authors use the term "threatened preterm birth" instead of "preterm labor"?2.Please add some references about definition of "threatened preterm birth"?
---

	-Inclusion criteria  1. Why do the researchers use the GA range only between 30-34 weeks? 2. It seems to have two groups of hospital that participate to this study. First is the hospital that uses CL which will use the criteria 1 or 2 and another is without CL which will use criteria 3 or 4. In my opinion, the positive fibronectin test has high false positive rate for preterm birth diagnosis. Please provides some reference that it can be used instead of 1,2 for diagnosis of preterm labor. 4. Multiple pregnancy and ruptured membrane have different outcomes and prognosis in many previous studies such as the study about progesterone for prevent preterm birth. What is the reason to included in this study? 5. Please mention about the frequency of uterine contraction. -Exclusion criteria  1. Is the advance cervical progression included in the criteria for exclusion? 2. Please mention about the initial assessment for exclude intra-uterine infection or fetal distress. -Intervention  1. Please add more details about the criteria for stopping tocolysis such as advanced cervical progression. and in case of cervical progression or increasing of uterine contraction. Will the protocol continue? 2. Please add more detail about "How often will you monitor the contraction or pelvic examination? Do you have the criteria for failure of inhibit labor? When will the patient can go home? How often will they follow up? What happen if the patient have contraction again after this protocol or remission after discharge? " 3. Please add more detail about MgSO₄ for neuroprotection which has the positive evidence for the preterm newborn. Is it used in this study? Is it affected the outcome of study? Statistics  1. Please add more detail about the reason for GEEs use in this study. The reviewer provided a marked copy with additional comments. Please contact the publisher for full details.
--	---

REVIEWER	Professor Ronald F Lamont BSc MB ChB DM FRCOG 1Research Unit of Gynaecology and Obstetrics, Department of Gynaecology and Obstetrics, Institute of Clinical Research, Odense University Hospital, University of Southern Denmark, Odense, Denmark. 2Division of Surgery, Northwick Park Institute of Medical Research Campus, University College London, London, UK. In the past, I advised or given lectures on conferences organised or supported by Glaxo-Smith-Klein, Sanofi-Synthelabo and Ferring Pharmaceuticals pertaining to PTB in general and tocolytics in particular. Currently I am and have been a member of an Independent Drug Monitoring Committee for randomised controlled trials of tocolytic agents.
REVIEW RETURNED	29-Mar-2019

GENERAL COMMENTS	REVIEW CHECKLIST: Q2. Is the abstract accurate, balanced and complete?
--

Response: Like many papers and grants I am asked to review, there is often a statement that although tocolytics delay birth, they have never been shown to improve neonatal outcome. This is the case in both the abstract and the introduction of this protocol. It should be recognized that no tocolytic study has ever been sufficiently powered to demonstrate such an effect.

Q3. Is the study design appropriate to answer the research question?

Response: Similar to my comments when I reviewed the APOSTEL III study for the Lancet, I have concerns about the inclusion of multiple pregnancy and those women with ruptured membranes. These two parameters muddy the waters since the mechanisms leading to preterm labour differ between singleton and multiple pregnancies, which may be relevant to the mode of action of tocolytics. The efficacy of tocolytics, also differs between singleton and twin pregnancies and between intact and ruptured membranes. When the sample size was calculated a priori were these two factors considered? The heterogeneity will be confusing and I'm not sure this will be corrected in the sensitivity analysis. Best to compare like with like and maintain standardisation: singleton pregnancies with intact membranes. In addition, atosiban is not licensed for use with ruptured membranes. If not excluded, these characteristics should be stratified in the randomisation to ensure roughly equal numbers in each group. Alternatively there should be a pre-specified subgroup analysis of singletons with intact membranes.

Q6. Are the outcomes clearly defined?

Response: I appreciate the need for composite outcomes when sample size precludes comparisons between rare complications. However, I have advocated the categorisation of neonatal complications into mild, moderate and severe to permit comparison. In this way, while it may not be possible to differentiate between the outcome of RDS between treatment groups, if this was subdivided in mild (head box oxygen only), moderate (CPAP) or severe (IPPV), there might be a differentiation. Similarly with infection: i) proven (positive blood cultures); ii) unproven (negative blood cultures but clinical signs of infection requiring a full course of antibiotics[Pen and Gent]) and iii) no infection (negative blood cultures and clinically well after 48h such that antibiotics can be discontinued). Perhaps this could be tested as part of the study.

Q8. Are the references up-to-date and appropriate?

Response 1: Care should be taken to qualify Reference No. 12, an update on the Cochrane Oxytocin Receptor Antagonist Review that has been severely criticised in feed back (only available to see on-line at the end of the document and may have been taken down by now). The review was criticized by a number of opinion leaders in the field. Prof Jim Thornton expressed concerns that there may be unintentional bias in favour of CCB due to: poor choice of outcomes reported in the abstract, which were better qualified in the text; different choice of reported outcomes compared with a previous CCB review by the same authors; poor objective and subjective judgment of trial quality; and poor choice of language showing a favorable opinion of CCBs. Prof Steve Thornton et al. were concerned that the methodology of the review did not enable the outcomes to be evaluated. They also felt that undue importance was attached to the rate of infant deaths from

one study, which was imbalanced at baseline and that the conclusion that CCBs were associated with a better neonatal outcome was not qualified. Prof Mats Åkerlund criticized the review for: having no rationale behind which trials were selected to evaluate both safety and efficacy; making conclusions about CCBs being superior to β -agonists when this was not the subject of the review; and exemplifying the drawbacks of meta-analysis with respect to risk of selection bias and absence of quality weighting. Prof Murph Goodwin, much of whose published research occupied the review, expressed his concerns that acknowledgement of his assistance implied that he concurred with their conclusion. He felt the analysis was 'flawed' and 'not up to the high standards of Cochrane reviews'. Other groups recorded that the review contained a worrying degree of subjectivism with respect to study inclusion and interpretation, which focused on data taken out of original context and without reference to subject profiles. These comments can be referenced by citing: Lyndrup J, and Lamont RF. The choice of a tocolytic for the treatment of preterm labor: a critical evaluation of nifedipine versus atosiban. *Expert Opin. Investig. Drugs* (2007) 16(6):843-853.

Response 2:

Surely there should be a reference to the Worldwide Comparative Trial of atosiban versus β 2-agonists, the largest randomized controlled trial of tocolytic therapy ever conducted. (Effectiveness and safety of the oxytocin antagonist atosiban versus beta-adrenergic agonists in the treatment of preterm labour. The Worldwide Atosiban versus Beta-agonists Study Group. *BJOG*. 2001;108(2):133-42).

Response 3:

Although atosiban is marketed as an oxytocin receptor antagonist to emphasise its uterospecificity, the drug's affinity for the vasopressin V1a receptor is many times greater than for the oxytocin receptor and so should really be known as a vasopressin/oxytocin receptor antagonist. (Lamont RF. The development and introduction of anti-oxytocic tocolytics. *BJOG*. 2003;110 Suppl 20:108-12).

OTHER COMMENTS:

- Why did the investigators restrict the gestational age to 30-34 weeks? Why did they not extend that down to 26 weeks albeit that would require stratification of the randomization?
- The investigators state that the use of tocolytics to maintain pregnancy may lead to intrauterine infection. What measures do they plan to use to test for this? (See response to Checklist Q6 above).
- Among the listed outcome measures:
 - For sepsis: i) the investigators say "culture proven sepsis". Do they mean blood cultures or swabs/gastric aspirate? I suggest they see response to Checklist Q6 above as a suggestion.
 - Checking for BPD at 36 weeks post-menstrual age seems rather early. The baby may only be 2-weeks old at this stage when it may

	be many more months before BPD becomes evident. The same applies to ROP and NEC.  • The outcome “time to birth” I assume means number of days gained? • Among maternal outcomes is listed prelabor rupture of the membranes. How can this be an outcome when ruptured membranes is an entry criterion? • Questionnaire follow up is suboptimal and will be criticized as in the ORACLE follow up study (2007) when data on cerebral palsy was obtained (and only in the UK and not Ireland) by a telephone call to the mother. Is there no option of a neurobehavioural assessment by a trained healthcare worker? • Is there really any need for maternal quality of life follow up at 3-months. Would the cost not be better directed to an infant neurobehavioural assessment by a trained healthcare worker. Among the subgroup analyses planned is “previous preterm birth”. Can they ensure that this relates to spontaneous preterm labour leading to preterm birth <34 weeks gestation, rather than slightly early physiological term labour at 36 completed weeks of gestation or elective, indicated preterm birth?
--	--

VERSION 1 – AUTHOR RESPONSE

Reviewer: 1

Reviewer Name

Tabata Zumpano Dias

Institution and Country

University of Campinas, Brazil

Please state any competing interests or state ‘None declared’:

None declared

Please leave your comments for the authors below

It was defined threatened preterm birth (PTB) as women with contractions and 4 strong risk factors for premature delivery. I understand these 4 risk factors as strong risk factors for premature delivery within 7 or 14 days but not necessarily 48 hours, that is the time that tocolysis is performed to allow the complete corticosteroid cycle.

As an obstetrician and researcher of spontaneous prematurity (my PhD was on tocolysis and corticosteroid in preterm labor in the largest study on prematurity in my country) I do not see why performing tocolysis if the woman has her cervix closed or if there is no modification of the cervix. I believe that many of the cases that could be included in the sample are cases that would benefit from the corticosteroid without tocolysis. In this way I think that the evaluation of the dilation / modification of the cervix is fundamental to define the need to use tocolysis. Without this evaluation of cervix I think the study does not make sense, so my option to reject (would be a change in methodology), despite understanding how important studies like this (randomized placebo controlled trials) are, mainly with a topic that is controversial (tocolysis) .

Thank you for your remarks. We agree that the main problem of tocolytic studies is diagnostics of preterm birth. The clinical findings of threatened preterm birth are poorly predictive of the diagnosis, so over-diagnosis is common until labor is well established.

Cervical dilatation is one of the symptoms of threatened preterm birth, but it has a very high inter-observer variation. In general practice, cervical dilatation is regarded as a cervix <15 mm, so these

patients are included in our study.

The criteria for threatened preterm birth we state in our protocol are based on the results of the APOSTEL 1 trial (Van Baaren GJ, Vis JY, Wilms FF, et al. *Predictive value of cervical length measurement and fibronectin testing in threatened preterm labor. Obstet Gynecol. 2014;123(6):1185-92*). They have been evaluated in many studies, and are used in current practice worldwide as diagnostic criteria for threatened preterm birth. For example, the same inclusion criteria were used in the APOSTEL III study, comparing atosiban versus nifedipine (Van vliet EOG, Nijman TAJ, Schuit E, et al. *Nifedipine versus atosiban for threatened preterm birth (APOSTEL III): a multicentre, randomised controlled trial. Lancet. 2016;387(10033):2117-2124*). Fifty percent of the participants delivered within one week after inclusion. Compared to other tocolytic studies, this is a relatively high number, thus confirming the use of these inclusion criteria in order to include a high risk group. In addition, cervical length measurement in combination with a fFn test are recommended in the national guidelines on preterm birth of the Dutch Society of Obstetrics and Gynaecology and the Royal College of Obstetrics and Gynaecology, and thus represent current clinical practice in many countries. We have added paragraph to the inclusion criteria in the methods section to explain this.

Reviewer: 2

Reviewer Name

Metha Songthamwat

Institution and Country

Department of Obstetric and Gynecology, Udonthani Hospital, Udonthani, Thailand.

Please state any competing interests or state 'None declared':

I have no conflict of interest with this article.

Please leave your comments for the authors below

Thank you very much for the opportunity to review this valuable protocol. My comments are listed below and a detail's file is attached to the editors and author -Title 1. Why do the authors use the term "threatened preterm birth" instead of "preterm labor"?

We acknowledge the absence of a commonly used definition in literature.

We reason that a period of threatened preterm birth is not always followed by labor/birth/delivery itself, therefore we do not call it 'labor', and we focus on the word "threatened". Synonyms can be "threatening" or "imminent" preterm birth.

Moreover, the term preterm birth is more in use in the United Kingdom than preterm labor.

2. Please add some references about definition of "threatened preterm birth"?

We refer to our answer to reviewer no. 1 (dr. Tabata Zumpano Dias)

We have added a paragraph to the inclusion criteria in the methods section to clarify this matter.

Inclusion criteria

1. Why do the researchers use the GA range only between 30-34 weeks?

We question the relevance of tocolysis in this group. Children born with 30-34 weeks of gestation have a relative good prognosis nowadays.

If the results of this study show no effect of tocolysis on the primary outcome, we plan to perform a study with women in threatened preterm birth GA 24-30 weeks subsequent to the APOSTEL 8 study.

2. It seems to have two groups of hospital that participate to this study. First is the hospital that uses CL which will use the criteria 1 or 2 and another is without CL which will use criteria 3 or 4. In my opinion, the positive fibronectin test has high false positive rate for preterm birth diagnosis. Please provides some reference that it can be used instead of 1,2 for diagnosis of preterm labor. It's true that the UK sites who participate have a different work up of threatened preterm birth than most Dutch Hospitals. In many UK hospitals, cervical length measurement is not part of local protocol, but the diagnosis is made on the combination of clinical findings and a positive fibronectine/Partus test. The Royal College of Obstetricians and Gynaecologists (RCOG) states in their guidelines on Preterm labour and birth, point 1.7.5 : *If fetal fibronectin testing is positive (concentration more than 50 ng/ml), view the woman as being in diagnosed preterm labour and offer treatment.*

The APOSTEL 8 study is a pragmatic study in which we aim to come as close to current clinical practice as possible in order to have the results of the study be applicable to the current situation in our hospitals

4. Multiple pregnancy and ruptured membrane have different outcomes and prognosis in many previous studies such as the study about progesterone for prevent preterm birth. What is the reason to included in this study?

It was our intention to design a pragmatic study by including all risk groups for preterm birth. Both women with multiple pregnancies and women with ruptured membranes are treated with tocolysis in the Netherlands, so they will also be included in this study.

We intend to perform a subgroup analysis using these predefined subgroups. We refer to our answer to Question 3 by reviewer no. 3 (Professor Ronald F Lamont)

5. Please mention about the frequency of uterine contraction.

We did not define the frequency of uterine contractions, as it has a very high inter-observer variation. If we name the frequency of contractions too strict, the results will not be representative for the whole population with threatened preterm birth (e.g. those with minimal contractions but who do develop cervical effacement).

Exclusion criteria

1. Is the advance cervical progression included in the criteria for exclusion?

No, there is no maximum limit.

2. Please mention about the initial assessment for exclude intra-uterine infection or fetal distress.

Intra-uterine infection and fetal distress are clinical diagnoses. It is left to the attending physician to make this diagnosis.

We think it is important to note that if we maintain strict definitions, the results will not be representative for the general population with threatened preterm birth. The same goes for the care during the intervention and the follow-up: we want to simulate a normal clinical setting, so the rest of the care should be according to local standards.

Intervention

1. Please add more details about the criteria for stopping tocolysis such as advanced cervical progression. and in case of cervical progression or increasing of uterine contraction. Will the protocol continue?

The same answer as question 2: continue unless the physician decides to stop, for example full dilatation or fetal distress.

2. Please add more detail about "How often will you monitor the contraction or pelvic examination? Do you have the criteria for failure of inhibit labor?"

Monitoring of contractions and pelvic examination will be assessed according to local protocol. We maintain strict in- and exclusion criteria, but the rest of the care should be as standard as possible.

When will the patient can go home? How often will they follow up? What happen if the patient have contraction again after this protocol or remission after discharge? "

Hospital discharge and follow-up will again be according to local protocol/attending physician.

If a patient develops a second period of threatened preterm birth, they will not be treated with a new course of corticosteroids and tocolysis, according to general practice worldwide.

3. Please add more detail about MgSO₄ for neuroprotection which has the positive evidence for the preterm newborn. Is it used in this study? Is it affected the outcome of study?

The use of MgSO₄ for neuroprotection will be up to the attending physician. Guidelines in the Netherlands and the UK state to offer MgSO₄ to woman with threatened preterm birth with GA < 30 weeks, and to *consider* offering to woman with GA 30-34 weeks. The use of MgSO₄ will be reported in the electronic Case Report Form (CRF)

Statistics

1. Please add more detail about the reason for GEEs use in this study.

We have added to the manuscript that GEEs will be used to take into account correlated outcomes in multiples.

Reviewer: 3

Reviewer Name

Professor Ronald F Lamont BSc MB ChB DM FRCOG

Institution and Country

1Research Unit of Gynaecology and Obstetrics, Department of Gynaecology and Obstetrics, Institute of Clinical Research, Odense University Hospital, University of Southern Denmark, Odense, Denmark.

2Division of Surgery, Northwick Park Institute of Medical Research Campus, University College London, London, UK.

Please state any competing interests or state 'None declared':

In the past, I advised or given lectures on conferences organised or supported by Glaxo-Smith-Klein, Sanofi-Synthelabo and Ferring Pharmaceuticals pertaining to PTB in general and tocolytics in particular. Currently I am and have been a member of an Independent Drug Monitoring Committee for randomised controlled trials of tocolytic agents.

Please leave your comments for the authors below

REVIEW CHECKLIST:

Q2. Is the abstract accurate, balanced and complete?

Response: Like many papers and grants I am asked to review, there is often a statement that although tocolytics delay birth, they have never been shown to improve neonatal outcome. This is the case in both the abstract and the introduction of this protocol. It should be recognized that no tocolytic study has ever been sufficiently powered to demonstrate such an effect.

We completely agree with this point of view, and this is exactly the reason to perform this study. We added a sentence to the introduction, stating no study has ever been sufficiently powered to demonstrate such an effect.

Q3. Is the study design appropriate to answer the research question?

Response: Similar to my comments when I reviewed the APOSTEL III study for the Lancet, I have concerns about the inclusion of multiple pregnancy and those women with ruptured membranes. These two parameters muddy the waters since the mechanisms leading to preterm labour differ between singleton and multiple pregnancies, which may be relevant to the mode of action of tocolytics. The efficacy of tocolytics, also differs between singleton and twin pregnancies and between intact and ruptured membranes. When the sample size was calculated a priori were these two factors considered? The heterogeneity will be confusing and I'm not sure this will be corrected in the sensitivity analysis. Best to compare like with like and maintain standardisation: singleton pregnancies with intact membranes. In addition, atosiban is not licensed for use with ruptured membranes. If not excluded, these characteristics should be stratified in the randomisation to ensure roughly equal numbers in each group. Alternatively there should be a pre-specified subgroup analysis of singletons with intact membranes.

During the design of the APOSTEL studies, this discussion was performed many times within the project group. We decided to design the studies as pragmatic studies that mimic the current clinical situation, in which women in threatened preterm birth pregnant with a multiple pregnancy are being treated analogous to singleton pregnancies. The results will therefore be important to the current clinical practice and subgroup analyses will be performed on singletons and multiples, as well as whether the membranes were intact, however we note that subgroup analyses are by definition underpowered. Nonetheless we did consider the impact of including these subgroups on the sample size as well as the feasibility of the study. The composite neonatal outcome is a clinically relevant one which only includes severe morbidity and mortality. This is the outcome that only expected to occur in 6% in the atosiban group. When the prevalence of the outcome is lower, the sample size is higher. Because adverse neonatal outcome occurs more often in multiples and ruptured membranes groups, keeping them in helps keep the sample size feasible. In the APOSTEL III study, a third of the

participants would likely have PPROM and a sixth would be multiple pregnancies, so including them also helps the feasibility.

We recognize the fact that atosiban is not licensed for woman with ruptured membranes. However, in many countries, such as the Netherlands and the UK, it is commonly used in women with ruptured membranes without signs of overt clinical infection. In addition, we note that other tocolytics are not registered but commonly used.

Q6. Are the outcomes clearly defined?

Response: I appreciate the need for composite outcomes when sample size precludes comparisons between rare complications. However, I have advocated the categorisation of neonatal complications into mild, moderate and severe to permit comparison. In this way, while it may not be possible to differentiate between the outcome of RDS between treatment groups, if this was subdivided in mild (head box oxygen only), moderate (CPAP) or severe (IPPV), there might be a differentiation. Similarly with infection: i) proven (positive blood cultures); ii) unproven (negative blood cultures but clinical signs of infection requiring a full course of antibiotics[Pen and Gent]) and iii) no infection (negative blood cultures and clinically well after 48h such that antibiotics can be discontinued). Perhaps this could be tested as part of the study.

As primary outcome, we choose to only include the most severe neonatal morbidity. These diseases have a low incidence in our population (GA>30 weeks), but come with a high risk of long-term consequences.

The other outcomes will be recorded in the CRF (e.g. for respiratory problems, we ask for: invasive mechanical ventilation, respiratory support, IRDS, pneumothorax). They will be analyses and reported as secondary outcomes.

Q8. Are the references up-to-date and appropriate?

Response 1: Care should be taken to qualify Reference No. 12, an update on the Cochrane Oxytocin Receptor Antagonist Review that has been severely criticised in feed back (only available to see on-line at the end of the document and may have been taken down by now). The review was criticized by a number of opinion leaders in the field. Prof Jim Thornton expressed concerns that there may be unintentional bias in favour of CCB due to: poor choice of outcomes reported in the abstract, which were better qualified in the text; different choice of reported outcomes compared with a previous CCB review by the same authors; poor objective and subjective judgment of trial quality; and poor choice of language showing a favorable opinion of CCBs. Prof Steve Thornton et al. were concerned that the methodology of the review did not enable the outcomes to be evaluated. They also felt that undue importance was attached to the rate of infant deaths from one study, which was imbalanced at baseline and that the conclusion that CCBs were associated with a better neonatal outcome was not qualified. Prof Mats Åkerlund criticized the review for: having no rationale behind which trials were selected to evaluate both safety and efficacy; making conclusions about CCBs being superior to β -agonists when this was not the subject of the review; and exemplifying the drawbacks of meta-analysis with respect to risk of selection bias and absence of quality weighting. Prof Murph Goodwin, much of whose published research occupied the review, expressed his concerns that acknowledgement of his assistance implied that he concurred with their conclusion. He felt the analysis was 'flawed' and 'not up to the high standards of Cochrane reviews'. Other groups recorded that the review contained a worrying degree of subjectivism with respect to study inclusion and interpretation, which focused on data taken out of original context and without reference to subject profiles. These comments can be referenced by citing: Lyndrup J, and Lamont RF The choice of a tocolytic for the treatment of preterm labor: a critical evaluation of nifedipine versus atosiban. Expert Opin. Investig. Drugs (2007) 16(6):843-853.

We too realize that the Cochrane review was of questionable quality, due to a variety of reasons. Overall, the evidence supporting tocolysis in placebo controlled studies (both nifedipine and atosiban) is very scarce. We have added this information to the introduction (sentence with reference no.12).

Response 2:

Surely there should be a reference to the Worldwide Comparative Trial of atosiban versus β 2-agonists, the largest randomized controlled trial of tocolytic therapy ever conducted. (Effectiveness and safety of the oxytocin antagonist atosiban versus beta-adrenergic agonists in the treatment of preterm labour. The Worldwide Atosiban versus Beta-agonists Study Group. BJOG. 2001;108(2):133-42).

We thank the reviewer to make this remark. Initially we did not include this reference in our introduction since ritodrine is no longer used in current practice in the Netherlands and the UK (partly

thanks to this study!). However, we agree that it's relevant to show that atosiban is effective in delaying birth.

We have added a sentence to the introduction, referring to the study.

Response 3:

Although atosiban is marketed as an oxytocin receptor antagonist to emphasise its uterospecificity, the drug's affinity for the vasopressin V1a receptor is many times greater than for the oxytocin receptor and so should really be known as a vasopressin/oxytocin receptor antagonist. (Lamont RF. The development and introduction of anti-oxytocic tocolytics. BJOG. 2003;110 Suppl 20:108-12). In literature, atosiban is mentioned as a oxytocin receptor antagonist, although it is known that it also has vasopressin receptor antagonist properties. We decided to maintain the classification of oxytocin receptor antagonist, since this is of most clinical relevance in this trial.

OTHER COMMENTS:

•Why did the investigators restrict the gestational age to 30-34 weeks? Why did they not extend that down to 26 weeks albeit that would require stratification of the randomization?

We refer to the answer mentioned above (question 3 by Metha Songthamwat):

We question the relevance of tocolysis in this group. Children born with 30-34 weeks of gestation have a relative good prognosis nowadays.

If the results of this study show no effect of tocolysis on the primary outcome, we will certainly perform following study with mothers GA 24-30 weeks.

•The investigators state that the use of tocolytics to maintain pregnancy may lead to intrauterine infection. What measures do they plan to use to test for this? (See response to Checklist Q6 above). No extra diagnostics tests shall be performed on mother or neonate apart from standard care, to simulate a normal clinical situation.

Outcomes for infection will be recorded in the Case Report Form:

- Intra-uterine infection
- Intra-partum antibiotics usage and indication
- If applicable, pathology assessment of the placenta
- Neonate: culture proven sepsis
- Neonate: meningitis

•Among the listed outcome measures:

•For sepsis: i) the investigators say "culture proven sepsis". Do they mean blood cultures or swabs/gastric aspirate? I suggest they see response to Checklist Q6 above as a suggestion. Culture proven sepsis is defined as a positive blood culture

•Checking for BPD at 36 weeks post-menstrual age seems rather early. The baby may only be 2-weeks old at this stage when it may be many more months before BPD becomes evident. The same applies to ROP and NEC. This is accounted for in the definition of BPD, see table 1 below. The primary point of assessment will not be 36 weeks post-menstrual age, but discharge from hospital. We removed this from the text.

TABLE 1. DEFINITION OF BRONCHOPULMONARY DYSPLASIA: DIAGNOSTIC CRITERIA

Gestational Age	< 32 wk	≥ 32 wk
Time point of assessment	36 wk PMA or discharge to home, whichever comes first	> 28 d but < 56 d postnatal age or discharge to home, whichever comes first
	Treatment with oxygen > 21% for at least 28 d plus	
Mild BPD	Breathing room air at 36 wk PMA or discharge, whichever comes first	Breathing room air by 56 d postnatal age or discharge, whichever comes first
Moderate BPD	Need* for < 30% oxygen at 36 wk PMA or discharge, whichever comes first	Need* for < 30% oxygen at 56 d postnatal age or discharge, whichever comes first
Severe BPD	Need* for ≥ 30% oxygen and/or positive pressure, (PPV or NCPAP) at 36 wk PMA or discharge, whichever comes first	Need* for ≥ 30% oxygen and/or positive pressure (PPV or NCPAP) at 56 d postnatal age or discharge, whichever comes first

Definition of abbreviations: BPD = bronchopulmonary dysplasia; NCPAP = nasal continuous positive airway pressure; PMA = postmenstrual age; PPV = positive-pressure ventilation.

•The outcome “time to birth” I assume means number of days gained?
Yes, from time of inclusion (baseline) to birth.

•Among maternal outcomes is listed prelabor rupture of the membranes. How can this be an outcome when ruptured membranes is an entry criterion?

Correct, this should have been removed. We thank the reviewer for noticing the error in the manuscript. In the current research protocol which is now in use, it was already removed.

•Questionnaire follow up is suboptimal and will be criticized as in the ORACLE follow up study (2007) when data on cerebral palsy was obtained (and only in the UK and not Ireland) by a telephone call to the mother. Is there no option of a neurobehavioural assessment by a trained healthcare worker?

The 3 month postpartum questionnaire (or actually 3 month after the due date) only involves questions about the Quality of Life and cost-effectiveness. The follow-up of children born prematurely is performed according to local protocol, which usually involves an out-patient clinic visit at 3 months corrected age.

We strive to perform a full neurodevelopmental and behavioral assessment at 2 and 5 years of age.

•Is there really any need for maternal quality of life follow up at 3-months. Would the cost not be better directed to an infant neurobehavioural assessment by a trained healthcare worker.

Yes, we argue that both maternal quality of life and infant neurodevelopment and behavioral are essential outcomes. The first is easy to assess using standardized questionnaires. The latter will be planned in a later stadium.

Among the subgroup analyses planned is “previous preterm birth”. Can they ensure that this relates to spontaneous preterm labour leading to preterm birth <34 weeks gestation, rather than slightly early physiological term labour at 36 completed weeks of gestation or elective, indicated preterm birth?

Yes, we can ensure this relates to spontaneous preterm birth only, since this is recorded in the Case Report Form. The question is formulated as following: *Previous SPONTANEOUS preterm birth (<37 weeks)?*

VERSION 2 – REVIEW

REVIEWER	Metha Songthamwat Department of Obstetrics and Gynecology, Udonthani Hospital. Udonthani, Thailand, 41000
REVIEW RETURNED	06-Aug-2019

GENERAL COMMENTS	The authors revised some parts of manuscript according to reviewer’s suggestion, however many questions are not be responded.  -The reason of only 30-34 weeks GA is used, was not explained. (The Apostel 3 study was on 25-34 weeks GA) -The authors mentioned “ the inclusion criteria are based on the results of the APOSTEL I study. Moreover, our previous APOSTEL III study showed that half of the women with these criteria deliver within seven days” (line 145-147) - but Apostel I was done on “Women with symptoms of preterm labor between 24 and 34 weeks, intact membranes, cervical length between 10 and 30 mm, and negative fibronectin test” and- Apostel III used the criteria “Threatened preterm birth was defined as at least three uterine contractions per 30 min and presence of one of the following: cervical length of 10 mm or less, both a cervical length of 11–30 mm and a positive fetal fibronectin test, or presence of ruptured amniotic membranes “ which is different from this study. Please explain more about the inclusion criteria. -I do not agree with the inclusion of multiple pregnancy and ruptured membrane group in this study for subgroup analysis, small number of multiple pregnancy cases will not have enough power for answer the question about atosiban in that specific groups. -The protocol of follow up and treatment in case of readmission (tocolysis) were not mentioned. -MgSO4 is effective as neuroprotective agent for preterm baby, however the tocolytic effect is also mentioned. It should be included in the associated factor to compare between groups.
---

VERSION 2 – AUTHOR RESPONSE

Reviewer 2:

The authors revised some parts of manuscript according to reviewer’s suggestion, however many questions are not be responded.

-The reason of only 30-34 weeks GA is used, was not explained. (The Apostel 3 study was on 25-34 weeks GA)

Answer: We apologize to the reviewer that we did not thoroughly explain this issue.

Yes, we agree that the effectiveness of tocolysis should be investigated in the whole group of women presenting with threatened preterm birth from GA 24-34 weeks.

Tocolysis is part of routine hospital care, and most obstetricians are convinced it has positive effect on the neonate, although there is no such evidence. Prior to this study, we undertook a national survey, asking obstetricians whether they would be willing to participate in a trial with a 50 % chance to randomize their patients in threatened preterm birth to a placebo. Many doctors answered they would be very reluctant to not give tocolysis to the very premature gestational ages (<30 weeks), so they were hesitant to participate in a study randomizing women under 30 weeks of gestation. The survey showed more willingness to investigate the effectiveness of tocolysis in women with threatened preterm birth between 30-34 weeks, therefore, we decided upon a feasible study design with eligibility above 30 weeks.

In addition, our project group questions the positive effect of tocolysis on the child's outcome, especially in middle/late preterm births (30-34 weeks). This group has a favorable prognosis, especially in the light of improved neonatal care and introduction of surfactant use, so we hypothesize that tocolysis as an intervention has no beneficial effect.

As we have stated before: If the results of this study show no effect of tocolysis on the primary outcome, we plan to perform a study with women in threatened preterm birth GA 24-30 weeks subsequent to the APOSTEL 8 study. In our opinion, however, the timing of a study was premature to include very preterm births in this study.

-The authors mentioned "the inclusion criteria are based on the results of the APOSTEL I study. Moreover, our previous APOSTEL III study showed that half of the women with these criteria deliver within seven days" (line 145-147).- but Apostel I was done on "Women with symptoms of preterm labor between 24 and 34 weeks, intact membranes, cervical length between 10 and 30 mm, and negative fibronectin test" and-Apostel III used the criteria "Threatened preterm birth was defined as at least three uterine contractions per 30 min and presence of one of the following: cervical length of 10 mm or less, both a cervical length of 11–30 mm and a positive fetal fibronectin test, or presence of ruptured amniotic membranes " which is different from this study. Please explain more about the inclusion criteria.

Answer: The APOSTEL I study is a diagnostic study, trying to find the best criteria for defining threatened preterm birth. The inclusion criteria for the APOSTEL 8 do not match the inclusion criteria of the APOSTEL I study, but are based on the results and conclusions of the APOSTEL I study. They are slightly different than they inclusion criteria used the APOSTEL III: During the set-up of the APOSTEL III trial, the results of the APOSTEL I weren't available yet.

(Note: two articles were written on the APOSTEL I cohort. We base our criteria on the diagnostic study: Van Baaren GJ, Vis JY, Wilms FF, et al. Predictive value of cervical length measurement and fibronectin testing in threatened preterm labor. *Obstet Gynecol.* 2014;123(6):1185-92).

We concluded from the APOSTEL I that women have a high risk of preterm birth with a cervical length <15 mm, or a cervical length 15-30 mm with a positive fibronectin test. These criteria have been evaluated in many studies, and are used in current practice worldwide as diagnostic criteria for threatened preterm birth.

The situation is a little different for sites where cervical length measurement is not part of local protocol (mainly United Kingdom), where the diagnosis is made on the combination of clinical findings and a positive fibronectine/Partus test. Cervical length measurement in combination with a fFn test are recommended in the national guidelines on preterm birth of the Dutch Society of Obstetrics and Gynaecology and the Royal College of Obstetrics and Gynaecology, and thus represent current clinical practice in many countries.

We have clarified this point by altering the protocol:

'These inclusion criteria are based on the results and conclusions of the APOSTEL I study and current guidelines within the Netherlands and UK. Moreover, our previous APOSTEL III study, with resembling inclusion criteria, showed that half of the women with these criteria deliver within seven days,10 validating this definition of women at high risk for preterm birth. In addition, the sample size of expected adverse neonatal outcome in the gestational age group of 30-34 weeks, was calculated from the APOSTEL III study.'

Concerning the frequency of contraction: we did not define the frequency of uterine contractions, as it has a very high inter-observer variation. If we specify the frequency of contractions too strict, the results will not be representative for the whole population with threatened preterm birth.

-I do not agree with the inclusion of multiple pregnancy and ruptured membrane group in this study for subgroup analysis, small number of multiple pregnancy cases will not have enough power for answer the question about atosiban in that specific groups .

Answer: We recognize this important issue and agree on many points of the reviewer. During the design of the APOSTEL studies, the discussion on eligibility of different groups of women was

performed thoroughly within the project group. We decided to design the studies as pragmatic studies that mimic the current clinical situation. Our sample size was calculated on the basis of the current clinical situation, in which singletons, multiples and women with ruptured membranes are being administered tocolysis and corticosteroids. These data were retrieved from our previous APOSTEL III study, in which these subgroups of patients were also included. The results will therefore be important to the current clinical practice and pre-specified subgroup analyses will be performed on singletons and multiples, as well as women with intact versus ruptured membranes. This section can be found in paragraph 'subgroup analysis' (line 269 of the manuscript). However we note and agree with the reviewer that subgroup analyses are by definition underpowered. The composite neonatal outcome is a clinically relevant outcome which only includes severe morbidity and mortality. This outcome is expected to occur in 6% in the atosiban group. Excluding women with ruptured membranes and multiple pregnancies would result in a lower prevalence of the primary outcome and a larger sample size, thus decreasing feasibility. In our previous APOSTEL III study, one third of the participants had ruptured membranes and one sixth of the included patients were women with multiple pregnancies. In order to clarify this important issue raised by reviewer 3, we have added the following section to the Methods section in the paragraph participants/eligibility criteria:

'In addition, the sample size of expected adverse neonatal outcome in the gestational age group of 30-34 weeks, was calculated from the APOSTEL III study.

This study was designed in a pragmatic fashion, in order for the results to be applicable in the current clinical practice. As most national guidelines and local protocols propose treatment for threatened preterm birth in both singleton and multiple pregnancies, as well as women with ruptured membranes, all these categories of patients are eligible for the study. '

-The protocol of follow up and treatment in case of readmission (tocolysis) were not mentioned.

Answer: Correct, we did not mention this because it should be done according to local protocol.

Readmission in case of a second episode of threatened preterm birth after initial inclusion in our study could be possible, but these women should not receive a second course of corticosteroids or a second course of tocolysis, according to international guidelines.

In case women did receive corticosteroids in current pregnancy during a previous admission, they cannot participate (an exclusion criterium).

-MgSO₄ is effective as neuroprotective agent for preterm baby, however the tocolytic effect is also mentioned. It should be included in the associated factor to compare between groups.

Answer: We acknowledge the fact that MgSO₄ is sometimes still regarded as a tocolytic drug, despite its proven ineffectiveness in the most recent Cochrane review. However, as it is effective as a neuroprotective agent in threatened preterm birth, it is allowed within the protocol to administer MgSO₄ for this purpose. Guidelines in the Netherlands and the UK state to offer MgSO₄ to woman with threatened preterm birth with GA<30 weeks, and to consider offering to woman with GA 30-34 weeks. However, MgSO₄ treatment > 30 weeks of GA is limited in the Netherlands. The total amount of women receiving MgSO₄ is expected to be quite low.

The use of MgSO₄ will be recorded in the electronic Case Report Form (CRF) and we will report on the percentage of women receiving MgSO₄. This has been added to the paragraph 'Interventions' in line 177.

Because the APOSTEL 8 is an randomized trial, we expect the amount of woman treated with MgSO₄ to be the same in both arms. Therefore, no stratification is needed.